# Fear of Cancer Recurrence in Patients with Sarcoma in the United Kingdom

**DOI:** 10.3390/cancers15030956

**Published:** 2023-02-02

**Authors:** Anika Petrella, Lesley Storey, Nicholas J. Hulbert-Williams, Lorna A. Fern, Maria Lawal, Craig Gerrand, Rachael Windsor, Julie Woodford, Jennie Bradley, Hatty O’Sullivan, Mary Wells, Rachel M. Taylor

**Affiliations:** 1Cancer Clinical Trials Unit, University College London Hospitals NHS Foundation Trust, London NW1 2PG, UK; 2Department of Psychology, Anglia Ruskin University, Cambridge CB1 1PT, UK; 3Department of Psychology, Edge Hill University, Ormskirk L39 4QP, UK; 4Patient Representative, University College London Hospitals NHS Foundation Trust, London NW1 2PG, UK; 5Sarcoma Unit, The Royal National Orthopaedic Hospital, Stanmore HA7 4LP, UK; 6Paediatric Directorate, University College London Hospitals NHS Foundation Trust, London NW1 2PG, UK; 7Iqvia, Ltd., Reading RG1 3JH, UK; 8Nursing Directorate, Imperial College Healthcare NHS Foundation Trust, London W2 1NY, UK; 9Centre for Nurse, Midwife and Allied Health Profession Research (CNMAR), University College London Hospitals NHS Foundation Trust, London NW1 2PG, UK

**Keywords:** sarcoma, fear of recurrence, psychological flexibility, distress, psychological impact

## Abstract

**Simple Summary:**

After a cancer diagnosis, the fear that it could come back is one of the most difficult negative emotions to manage. Sarcoma is a rare cancer of connective tissue affecting soft tissue and bone that has a high rate of recurrence and metastases. It can present itself in any age group from childhood to older adulthood. The experience of fear of cancer recurrence has not yet been explored in-depth among those with sarcoma. We, therefore, conducted an online survey to identify the prevalence of fear of cancer recurrence and factors that may be associated with it. A total of 229 people with sarcoma submitted responses, and the majority expressed interest in receiving support for fear of cancer recurrence. Overall, fear of cancer recurrence levels was found to be higher than those reported by patients with most other types of cancer. Emotional distress and being able to manage emotions were associated with fear of cancer recurrence.

**Abstract:**

Fear of cancer recurrence (FCR) is a persistent concern among those living with cancer and is associated with a variety of negative psychosocial outcomes. However, people with sarcoma have been underrepresented within this area of research. We aimed to determine the prevalence of FCR experienced by people with sarcoma in the United Kingdom and explore factors that may predict FCR, such as the perceived impact of cancer and psychological flexibility. Participants (*n* = 229) with soft tissue (*n* = 167), bone (*n* = 25), and gastrointestinal stromal tumours (*n* = 33) completed an online survey including the self-reported measures of FCR, the perceived physical and psychological impact of cancer and psychological flexibility, and demographic information. Data were analysed using ANOVA and multiple regression modelling. Mean FCR scores (*M* = 91.4; *SD* = 26.5) were higher than those reported in meta-analytic data inclusive of all cancer types (*M* = 65.2; *SD* = 28.2). Interest in receiving support for FCR was also high (70%). Significant factors associated with FCR included cognitive and emotional distress and psychological flexibility, but not perceptions of the physical impact of cancer (*R^2^* = 0.56). The negative association between psychological flexibility and FCR suggests the potential benefit of intervention approaches which foster psychological flexibility, such as acceptance and commitment therapy.

## 1. Introduction

Fear of cancer recurrence (FCR) is a highly prevalent and distressing psychological challenge for those living with and beyond cancer [1] and is defined as “fear, worry, or concern that cancer may come back or progress” [2]. FCR is considered one of the most distressing consequences of cancer, demonstrating associations with impaired physical and psychosocial functioning and lower overall quality of life [3,4,5]. It is reported to occur in 39–97% of cancer survivors; the prevalence of FCR is dependent on how it is measured and the definition of clinical levels of FCR [6,7]. Managing FCR has been highlighted as the number one unmet need among cancer survivors [6]. It does not appear to dissipate with time and, if left unaddressed, can become a complex lifelong concern [4,5,6].

Research and understanding of FCR have grown rapidly [8]; however, uncertainties remain regarding prevalence by cancer type, underlying mechanisms of action, and the best practices for intervention [9]. As identified by the James Lind Alliance’s ‘Living with and beyond cancer’ research priorities, interventions to best support individuals to cope with FCR are needed [10]. People with sarcoma have been significantly underrepresented within FCR research. This is a concern given that recurrence rates are higher within the sarcoma population than most other solid cancers [11]. In addition, sarcomas are a rare and diverse group of cancers characterized by considerable clinical heterogeneity and significant physical burden on survivors [12]. Thus, sarcoma-specific research is warranted in order to inform intervention development and delivery.

To date, only two reports have included people with sarcoma in their sample [13,14]. First, an observational study in patients with gastrointestinal stromal tumors (GIST) examined fear of progression (not recurrence) and found that approximately 50% of patients had high levels of fear, which were associated with greater psychological distress [13]. Second, a network analysis of FCR among young adult cancer patients included a small number of people with sarcoma in their sample (*n* = 19; 7.7%) and reported that the majority of their sample scored above the established cut-off for high FCR on their measure (fear of progression questionnaire short form; FoP-Q-SF) [14]. Given the lack of evidence and the need for tailored interventions, additional work is required that is specific to the sarcoma population.

In the broader FCR literature, higher FCR scores have been associated with younger age, female gender, physical symptoms (e.g., pain, fatigue) and greater anxiety, and depression [8]. Potential underlying mechanisms have been explored and include optimism and social support [4,6]. These findings emphasize the potential interplay between the physical and psychological challenges faced by patients that may be impacting FCR and the need for evidence-based supportive interventions. Interventions that have been developed to date to manage FCR have included cognitive behavioral techniques, as well as relaxation, meditation, and other positive psychology-based approaches [15]. Acceptance and commitment therapy (ACT)-based interventions have shown promise in reducing cancer-related distress and FCR [16,17]. However, a greater understanding of specific active mechanisms or intervention components is needed in order to impact FCR for a range of subgroups.

Psychological flexibility is a core component of ACT [13] and may be crucial in understanding how individuals are affected by, and cope with, the significant challenges brought on by cancer and its treatments. Psychological flexibility is defined as the ability to identify and adapt to situational demands in an attempt to improve longer-term outcomes in a way that is personally meaningful [18,19]. It has been associated with improved psychological health, quality of life, and well-being in both clinical and non-clinical populations [20,21,22,23,24], including both distress-related and positive outcomes (e.g., benefit finding) in cancer survivors [25]. Furthermore, psychological flexibility has been shown to be amenable to change over time, presenting a potential target for interventions.

This study aimed to determine the prevalence of FCR among people with sarcoma in the United Kingdom (UK) and explore associated factors specific to the physical and psychological impacts of cancer and psychological flexibility.

## 2. Materials and Methods

### 2.1. Participants & Recruitment

Following approval from an institutional research ethics committee (Birmingham City University: Storey/#9678/sub2/R(A)/2021/Jul/BLSS FAEC), patients with sarcoma living in the UK were invited to participate in an online cross-sectional survey. The survey was administered by Quality Health using their in-house online survey software, which was open for 13 weeks (July to October 2021). Patients self-identified to participate after receiving information from sarcoma-specific and cancer charities via newsletters or social media posts. Patients were eligible to participate if they met the following criteria: diagnosis of sarcoma (any type); receiving all or some of their care in the UK; aged 16 or over; literate in English; and provided consent to participate (i.e., submitted survey was implicit of the consent). Confirmation of eligibility was required to proceed with the survey.

### 2.2. Measures

A bespoke survey was developed in collaboration with an established sarcoma patient advisory group and informed by previous work [26]. The survey included investigator-designed questions and validated measures of FCR, the perceived physical and psychological impact of cancer and psychological flexibility.

#### 2.2.1. Fear of Cancer Recurrence

The fear of cancer recurrence inventory (FCRI) is a 42-item scale, widely established, an in-depth measure of FCR [27]. A total score was obtained (ranging from 0 to 168), with higher scores indicating greater FCR. The FCRI has been utilized in clinical and research practice and has been shown to be valid and reliable [8].

#### 2.2.2. Activities of Daily Living

Two items from the Toronto extremity salvage score (TESS) [28] were used to measure perceived disability status and overall impact on activities of daily living (ADL). Both questions were answered on a 1–5 scale, with the ability to perform ADLs during the past week ranging from ‘not at all difficult’ to ‘impossible to do’ and self-reported disability status as ‘not at all disabled’ to ‘completely disabled’. The TESS is widely used as a patient-reported functional assessment following the diagnosis and treatment of upper and lower extremity sarcoma [29,30,31,32] and has been tested for validity and reliability [28]. A generic version combining both the upper and lower extremity scale was developed so it could be administered generically to patients with any type of cancer and contained 48 items.

#### 2.2.3. Psychology Impact

The psychological impact of cancer (PIC) scale is a valid and reliable tool for assessing the perceived psychological impact of cancer [33]. This scale contains 12 items answered on a scale of 1–4 ranging from ‘definitely does not apply to me’ to ‘definitely applies to me’, which then make up four individual subscales (cognitive distress, cognitive avoidance, fighting spirit, and emotional distress). Higher scores (ranging from 3 to 12) on each subscale represent the greater endorsement of the said factor (e.g., cognitive distress). The PIC has been validated in patients living with and beyond cancer in the UK and Australia [33].

#### 2.2.4. Psychological Flexibility

Psychological flexibility was assessed using the comprehensive assessment of acceptance and commitment therapy processes (CompACT) [34]. This measure is comprised 23 items assessing the key dyadic process of psychological flexibility scored on a 7-point Likert scale, ranging from 0 to 6 (‘strongly disagree’ to ‘strongly agree’). The total sum score ranges from 0 to 138, with higher scores indicating greater psychological flexibility. The CompACT has been shown to be valid and reliable within nonclinical populations [18,35,36], as well as within oncology settings [37].

Personal characteristics were collected to describe the sample and included gender, age, ethnicity, sexual orientation, marital status, employment status, and caregiver status. Cancer-specific characteristics were collected on the type of sarcoma, the year diagnosed, treatments received, amputation status, as well as a history of recurrence and/or metastatic disease. Interest in engaging with support specific to FCR was also queried.

### 2.3. Analysis

Data were analysed in R (Version 3.6.1). Data were inspected for missing values and then described: normally distributed data by the mean and standard deviation (SD), and binary and categorical variables were presented using frequency and percentages. The prevalence and magnitude of FCR in this sample were described and referenced in relation to available meta-analytic data and were inclusive of multiple cancer types reported in the literature. To explore differences in FCR by sarcoma type (soft tissue, bone, GIST), a one-way ANOVA was conducted, adjusting for multiple comparisons using the Tukey method. Bivariate correlations examined the size and direction of correlation between theoretically hypothesized associated factors. Multiple regression analysis was performed to establish how much variance in FCR scores was explained by physical and psychological impacts of cancer and psychological flexibility in this sample when accounting for relevant personal and cancer-specific characteristics [i.e., age, gender, marital status (single/coupled), time since diagnosis and recurrence status (no/yes)].

## 3. Results

In total, 229 people with sarcoma aged 18–85 completed the survey. Personal and cancer-specific characteristics are summarized in Table 1. The majority of respondents identified as female (*n* = 168, 73%), were married or in a long-term relationship (*n* = 165, 73%), employed (*n* = 133, 60%), and white (*n* = 216, 96%). Most participants had been diagnosed with soft tissue sarcoma (*n* = 165, 74%), received surgery (*n* = 177, 77%), and were not on active treatment (*n* = 189, 86%).

Bivariate correlations between study variables are displayed in Table 2. Results from the multiple linear regression analysis are summarized in Table 3. Of the covariates included less time since diagnosis and reporting not having had a recurrence were significantly associated with greater FCR. Specific to the psychological impact of cancer, cognitive distress, and emotional distress were positively and significantly associated with higher levels of FCR, whereas cognitive avoidance and fighting spirit were not significantly associated. Psychological flexibility was negatively and significantly associated with lower levels of FCR. No significant associations were found between perceptions of the physical impact of cancer (i.e., disability status and impact on ADL). The model accounted for 56% (95% CI = 0.42, 0.61) of the variance in FCR among people with sarcoma in our sample.

## 4. Discussion

In this study, we observed the prevalence of FCR among people with sarcoma in the UK, explored associated factors specific to the physical and psychological impact of cancer, and examined the role of psychological flexibility. Compared to the meta-analytic data of common cancer types [8], our study reported high levels of FCR. Participants expressed an interest in engaging in supportive interventions, which highlights the need for support among this population. Our findings demonstrate that the psychological impact of cancer, specifically cognitive and emotional distress, are significantly associated with greater levels of FCR, whereas the physical impact of cancer is insignificantly associated. Lastly, psychological flexibility was found to be negatively associated with FCR, representing a potential target for intervention development.

People with sarcoma have been historically underrepresented in FCR research. When examining the prevalence of FCR, mean scores reported across 10 different cancer types ranging from 39.8 among prostate cancer survivors to 113.5 among gynaecological cancer survivors, with standard deviations ranged from 18.6 to 28.2 using the FCRI [8]. The mean score among our sample of people with sarcoma was 91.4 (*SD* = 26.5). This is higher than the overall combined, weighted mean FCRI-Total score inclusive of all cancer types, which was reported as 65.2 (95% CI: 58.0–72.3) [8]. Only the estimate reported among the sample of gynaecological cancer patients [38] was higher than that observed in our sample. A higher FCR may be attributed to higher rates of recurrence in this population and the impact of sarcoma and its treatment on physical and psychological well-being and quality of life. Interestingly, our sample was predominantly female, in common with the gynaecological sample. Previous research has emphasised gender as a relevant demographic factor [4,6,7]; however, our own findings may be confounded by the unequal gender representation within our sample. Furthermore, a higher prevalence of anxiety has been noted in females [39]; thus, FCR, a form of state anxiety, may also be associated with differences in gender. Nonetheless, the reasons for gender-based differences within FCR severity have yet to be definitively identified and should be explored.

In addition to being identified as an unmet need among cancer survivors [6] and a research priority within the UK [10], FCR has been associated with increased costs to healthcare systems [40,41]. Thus, it is imperative that this field of research focuses on intervention development, testing, and optimization. Findings from this study support the need for interventions aimed at patients with sarcoma, given their high FCR. Furthermore, the high level of interest in supportive care interventions throughout the cancer care continuum highlights the demand for these interventions to be offered continuously starting from the time of diagnosis.

Recent efforts to develop interventions aimed at managing FCR have focused on a mind–body approach, which addresses the physical, cognitive, emotional, and behavioural aspects of the cancer experience [15]. Distress is a complex experience that results from the individual interplay of these aspects and is highly variable within and between individuals. The PIC scale used in this study to assess components of the psychological impact of cancer was developed using items previously forming the Mini-Mental Adjustment to Cancer Scale [42] to provide a brief and conceptually accessible tool with good psychometric properties. However, it is important to note that the psychometric properties of the fighting spirit sub-scale remain poor. In our sample, cognitive distress and emotional distress explained some of the unique variances of FCR. Thus, interventions should consider focusing on strategies that are aimed at reducing cognitive and emotional distress.

Psychological flexibility emerged as a factor associated with reduced FCR in our sample, highlighting the critical role this construct can play in facilitating psychological health and adjustment to cancer. Given the unique profile of people with sarcoma, it is essential that programs respond effectively to the challenges of this diagnosis in pursuit of mitigating long-term goals around health and wellbeing. This study is cross-sectional, so it does not provide any insight into the causality of associations; however, observations are in line with broader theory and evidence within this clinical population and provide justification for the continued exploration of this key construct.

Of the covariates included in our model, the time since diagnosis and recurrence status emerged as associated factors, with lower levels of FCR observed among those further from diagnosis, as well as those who had already experienced recurrence. The type of sarcoma also emerged as an associated factor, with higher levels of FCR observed in those with soft tissue sarcoma. Based on the prior literature [13,14] and clinical experience, it was hypothesised that marital status (as a form of support), age, and gender would account for some of the unique variances in FCR; however, this was not the case in these data. For example, recent work has demonstrated that younger patients may have greater severity of FCR compared to older patients [14]. It could be surmised that this is due to heightened levels of psychological distress in younger age groups, underdeveloped coping strategies, and concerns regarding developmental tasks related to the life stage at the time of diagnosis [14]. Given sample characteristics, findings specific to demographic and clinical characteristics may be due to a lack of representation and should be explored further.

### Limitations

There are limitations to consider when interpreting the findings of this study. This study was cross-sectional; thus, causal and temporal inferences are not possible. The sampling approach was open to self-selection bias in that access was limited to those who used participating charities or social media. We recognise that our sample was predominately white and female. However, there is a similar proportion to the types of sarcoma that are represented in the UK. Future work should aim for a more balanced distribution of demographic characteristics through a more targeted sampling strategy. A longitudinal design would allow for the exploration of temporal changes and opportunities to explore causality between the variables assessed in this study. Whilst the selection of variables included in our analytic models was theoretically driven and accounted for the majority of the variance in FCR, there are additional constructs yet to be identified that may provide additional insight into mechanisms of action in FCR. Despite the aforementioned limitations, this is the largest study reporting FCR in patients presenting with sarcoma, and findings from this study provide valuable insight into this understudied population.

## 5. Conclusions

To our knowledge, this is the first study to explore the prevalence of FCR among those living with and beyond sarcoma in the UK. In comparison to other cancer types, a high prevalence and severity of FCR were observed. The psychological impact of sarcoma and the potential benefit of fostering psychological flexibility when aiming to address FCR demonstrates the importance of addressing cognitive and emotional distress after a sarcoma diagnosis. Interventions targeting these constructs, for example, acceptance and commitment therapy-based approaches, warrant further investigation and hold promise for managing FCR in both the short- and long-term.

## Figures and Tables

**Table 1 cancers-15-00956-t001:** Patient participant personal and cancer specific characteristics.

Characteristic	Mean (*SD*)/*n* (%)
Age	52.45 (14.7)
Gender	
Male	61 (27%)
Female	168 (73%)
Ethnicity	
White	216 (96%)
Other	9 (4%)
Marital status	
Married/in long-term relationship	165 (73%)
In a relationship but not cohabitating	12 (5%)
Single	32 (14%)
Widowed or divorced	19 (8%)
Employment status	
Employed full time or part time	133 (60%)
Permanently sick/disabled	26 (12%)
Retired	61 (28%)
Caregiver status	
Yes	64 (29%)
No	153 (71%)
Type of sarcoma	
Soft tissue sarcoma	167 (74%)
Bone sarcoma	25 (11%)
GIST	33 (15%)
Time since diagnosis	
<1 year	14 (7%)
2 to 5 years	107 (51%)
6 to 10 years	57 (27%)
>10 years	31 (15%)
Treatments received *	
Surgery	177 (77%)
Radiotherapy	16 (7%)
Chemotherapy	28 (12%)
Other	21 (9%)
History of recurrence	
Yes	53 (24%)
No	149 (68%)
Unknown	17 (8%)
History of metastatic disease	
Yes	58 (26%)
No	146 (66%)
Unknown	17 (8%)
Amputation status	
Yes	18 (9%)
No	168 (79%)
Not applicable	26 (12%)
Disability status	
Not at all disabled	87 (45%)
Mildly to moderately disabled	69 (35%)
Severely or completely disabled	39 (20%)

* = multiple responses given; GIST: gastrointestinal stromal tumours.

**Table 2 cancers-15-00956-t002:** Bivariate correlations of study variables.

Variable	1	2	3	4	5	6	7	8	9	10	11	12
1. Age	-											
2. Gender	−0.07	-										
3. Marital status	−0.15	0.13 *	-									
4. Time since diagnosis	0.21 **	−0.04	0.00	-								
5. Recurrence status	0.17 *	0.10	−0.07	0.29 **	-							
6. Disability status	−0.07	−0.06	0.16 *	0.13	−0.02	-						
7. ADL impact	−0.09	−0.04	0.20 **	0.12	−0.02	0.72 **	-					
8. Cognitive distress	−0.09	0.03	0.09 *	−0.10	0.06	0.26 **	0.20 **	-				
9. Cognitive avoidance	0.04	0.09	0.09	0.01	0.07	−0.10	0.02	0.24	-			
10. Emotional distress	−0.19 **	0.22 **	−0.03	−0.16 *	0.12	0.24 **	0.27 **	0.65	0.19 **	-		
11. Fighting spirit	−0.12	0.01	−0.01	−0.06	0.05	0.20 **	0.23 **	0.10	0.24 **	0.27 **	-	
12. Psychological flexibility	0.27 **	−0.06	−0.17 *	0.12	0.10	−0.14 *	−0.24 **	−0.54	−0.22 **	−0.42 **	−0.03	-
13. FCR	−0.18 *	0.19 **	0.06	−0.23 **	0.17 *	0.20 **	0.19 **	0.57 **	0.15 *	0.69 **	0.25 **	−0.47 **

* Indicates *p* < 0.05 ** indicates *p* < 0.01; ADL: activities of daily living; FCR: fear of cancer recurrence.

**Table 3 cancers-15-00956-t003:** Regression results for FCR.

Factor	*b*	95% CI	*beta*	95% CI	*sr^2^*	95% CI
(Intercept)	63.85 **	[27.28, 100.43]				
Age	0.01	[−0.19, 0.22]	0.01	[−0.11, 0.13]	0.00	[−0.00, 0.00]
Gender	2.84	[−3.71, 9.40]	0.05	[−0.07, 0.17]	0.00	[−0.01, 0.01]
Marital status ^+^	−1.45	[−8.86, 5.96]	−0.02	[−0.14, 0.09]	0.00	[−0.00, 0.00]
Time since diagnosis	−0.63 *	[−1.21, −0.04]	−0.13	[−0.26, −0.01]	0.01	[−0.01, 0.04]
Recurrence status	−9.73 *	[−17.54, −1.93]	−0.15	[−0.27, −0.03]	0.02	[−0.01, 0.05]
Disability status	2.00	[−2.91, 6.91]	0.07	[−0.10, 0.23]	0.00	[−0.01, 0.01]
ADL impact	−2.03	[−6.39, 2.34]	−0.08	[−0.24, 0.09]	0.00	[−0.01, 0.01]
Cognitive distress	2.15 *	[0.26, 4.03]	0.19	[0.02, 0.35]	0.02	[−0.01, 0.04]
Cognitive avoidance	−0.51	[−1.95, 0.93]	−0.04	[−0.16, 0.08]	0.00	[−0.01, 0.01]
Emotional distress	5.39 **	[3.44, 7.34]	0.44	[0.28, 0.61]	0.09	[0.03, 0.15]
Fighting spirit	0.61	[−1.02, 2.23]	0.05	[−0.08, 0.18]	0.00	[−0.01, 0.01]
Psychological flexibility	−0.22 **	[−0.38, −0.05]	−0.19	[−0.33, −0.05]	0.02	[−0.01, 0.05]

Note. A significant *b*-weight indicates that the beta-weight and semi-partial correlation are also significant. *b* represents unstandardized regression weights. *beta* indicates the standardized regression weights. *sr^2^* represents the semi-partial correlation squared. *r* represents the zero-order correlation. Figures in brackets indicate the lower and upper limits of a confidence interval, respectively. ^+^ marital status was dichotomized to 1 = coupled; 2 = uncoupled (single, widowed, or divorced). * Indicates *p* < 0.05. ** indicates *p* < 0.01.

## Data Availability

Data are available on request from the authors.

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
