# Peer review of "Fear of Cancer Recurrence in Patients with Sarcoma in the United Kingdom"

_cancers, 2023, doi:10.3390/cancers15030956_

Round 1
Reviewer 1 Report
General comments:
The present manuscript deals with an important topic, which concerns patients suffering from sarcoma. The authors performed an interesting cross-sectional survey study among sarcoma survivors investigating the prevalence of fear of cancer recurrence (FCR)n experienced by people with sarcoma in the United Kingdom and explore factors which may predict FCR, such as perceived impact of cancer and psychological flexibility. The present manuscript is well written. It allows important insights in this topic. The methods were appropriate, and the results as well as the discussion are written in a clear manner. I have one question to an important variable.
Special comments:
Page 4:
Table 1.:
- Please discuss the role of caregivers concerning fear in recurrence
- Patients with local sarcoma as well advanced disease were included. Might metastases have an impact on FCR?
- What is about the factor time since diagnosis? Might this have influence FCR?
Author Response
The present manuscript deals with an important topic, which concerns patients suffering from sarcoma. The authors performed an interesting cross-sectional survey study among sarcoma survivors investigating the prevalence of fear of cancer recurrence (FCR) experienced by people with sarcoma in the United Kingdom and explore factors which may predict FCR, such as perceived impact of cancer and psychological flexibility. The present manuscript is well written. It allows important insights in this topic. The methods were appropriate, and the results as well as the discussion are written in a clear manner.
- Thank you for reviewing our manuscript and your comments.
I have one question to an important variable.
Special comments:
Page 4, Table 1.
- Please discuss the role of caregivers concerning fear in recurrence
- We included marital status as a variable in the regression analysis, which would reflect the role of caregivers. However, this was not significant. We included text in the discussion “it was hypothesised that marital status (as a form of support), age and gender would account for some of the unique variance in FCR, however this was not the case in these data.” As this is not significant, we do not feel it necessary to discuss further.
- Patients with local sarcoma as well advanced disease were included. Might metastases have an impact on FCR?
- We included history of metastases in our demographic data, which is not necessarily advanced disease in patients with sarcoma. We did not include this in the analysis as it is not something that has been reported previously. However, this warrants further exploration in the future.
- What is about the factor time since diagnosis? Might this have influence FCR?
- As noted in Tables 2 and 3, there was a significant correction between time since diagnosis and FCR and this was identified as one of the predicting factors.
Reviewer 2 Report
It was a true pleasure to read this well-written and organised manuscript. There was a good clarity of expression and a detailed account of handling of missing data. It is hard to identify many recommendations for improvement.
There are two minor comments as follows:
Line 68: Add detail after “reports” FCR or progression
Line 89: remove “a” before crucial.
One major issue relates to the highly selected nature of the participants and the lack of representation of certain demographic groups (males, non-white). We do not know how representative this sample is of people with sarcoma in the UK. This is no doubt a function of the sources of recruitment - social media and charities where people will be proactive, problem-focused in their coping strategy and potentially presenting with a greater level of concern (this might represent the reason for their engagement with the research study and also the opportunity to express that they need support). I would like to see further recognition of this.
I hope this feedback is of value.
Author Response
It was a true pleasure to read this well-written and organised manuscript. There was a good clarity of expression and a detailed account of handling of missing data. It is hard to identify many recommendations for improvement.
- Thank you for taking the time to review our paper and thank you for your comment.
There are two minor comments as follows:
Line 68: Add detail after “reports” FCR or progression
- The rest of this paragraph summarises the findings of these two reports so it is unclear what additional information the reviewer requires. However, we have added the references after the sentence for clarity.
Line 89: remove “a” before crucial.
- This has been removed as requested.
One major issue relates to the highly selected nature of the participants and the lack of representation of certain demographic groups (males, non-white). We do not know how representative this sample is of people with sarcoma in the UK. This is no doubt a function of the sources of recruitment - social media and charities where people will be proactive, problem-focused in their coping strategy and potentially presenting with a greater level of concern (this might represent the reason for their engagement with the research study and also the opportunity to express that they need support). I would like to see further recognition of this.
- This was acknowledged in the limitations section of the discussion, with recommendations for future research. If the reviewer requires additional text, we would be happy to add to this if they would clarify what in addition to existing text they think needs adding.